# Sheep Face Detection Based on an Improved RetinaFace Algorithm

**DOI:** 10.3390/ani13152458

**Published:** 2023-07-29

**Authors:** Jinye Hao, Hongming Zhang, Yamin Han, Jie Wu, Lixiang Zhou, Zhibo Luo, Yutong Du

**Affiliations:** College of Information Engineering, Northwest A&F University, Xianyang 712100, China; cocohao2001@126.com (J.H.); zhm@nwsuaf.edu.cn (H.Z.);

**Keywords:** sheep face detection, improved RetinaFace, lightweight, attention module, computer vision

## Abstract

**Simple Summary:**

Accurate farming is essential for optimal pasture management and productivity improvement. Recently, automatic sheep face detection has become a promising solution for accurate farming. In this study, we explore a novel sheep face detection scheme based on lightweight convolutional neural network. The results have shown that our proposed detection method has the characteristics of real-time and robust detection, which provides a potential solution for accurate sheep face detection on actual sheep farms.

**Abstract:**

The accurate breeding of individual sheep has shown outstanding effectiveness in food quality tracing, prevention of fake insurance claims, etc., for which sheep identification is the key to guaranteeing its high performance. As a promising solution, sheep identification based on sheep face detection has shown potential effectiveness in recent studies. Unfortunately, the performance of sheep face detection has still been a challenge due to diverse background illumination, sheep face angles and scales, etc. In this paper, an effective and lightweight sheep face detection method based on an improved RetinaFace algorithm is proposed. In order to achieve an accurate and real-time detection of sheep faces on actual sheep farms, the original RetinaFace algorithm is improved in two main aspects. Firstly, to accelerate the speed of multi-scale sheep face feature extraction, an improved MobileNetV3-large with a switchable atrous convolution is optimally used as the backbone network of the proposed algorithm. Secondly, the channel and spatial attention modules are added into the original detector module to highlight important facial features of the sheep. This helps obtain more discriminative sheep face features to mitigate against the challenges of diverse face angles and scale in sheep. The experimental results on our collected real-world scenarios have shown that the proposed method outperforms others with an *F*_1_
*score* of 95.25%, an average precision of 96.00%, a model size of 13.20 M, an average processing time of 26.83 ms, and a parameter of 3.20 M.

## 1. Introduction

With the increasing need for high-quality animal husbandry products such as milk and meat, accurate breeding of sheep has drawn increasing attention in animal husbandry studies, which can improve product outputs and quality traceability by automated and precise management of livestock [1,2]. Accurate sheep identification is one of the key techniques for accurate farming. Typically, there are many common methods that use GPS necklaces, ear tags, and radio frequency identification (RFID) for animal identification [3,4,5,6]. However, those devices are expensive, time-consuming, and even cause physical damage to livestock. What’s more, the performance of those methods is limited by the effective monitoring distance and device reliability, which do not meet the requirement of remote monitoring and precision livestock farming. Therefore, it is of great importance to design a more robust sheep identification scheme for modern accurate breeding.

Recently, with the rapid development of machine learning, computer vision, and deep learning, animal identification has shifted from manual devices to automation. Automatic sheep identification methods generally require the use of a vision system equipped with sensors, such as cameras, to monitor animal groups, and specific animals are identified through processing and analyzing sensor data based on computer vision techniques. These methods have the advantages of low cost and cause no harm to livestock. Early automatic animal identification mainly relied on traditional machine learning methods (e.g., SVM or KNN). For example, Kumar et al. [7] utilized a range of traditional feature extraction methods, such as independent-candid covariance-free incremental PCA (IND-CCPCA) combined with a support vector-based classifier for cattle face recognition. Shahinfar et al. [8] explored the feasibility of using three machine-learning algorithms to analyze dairy cow images. More recent methods [9,10,11] have been proposed to improve identification performance by utilizing novel convolutional neural network (CNN) models. For example, Li et al. [10] used lightweight CNN to recognize the individual identity of beef cattle. During the same period, Hitelman et al. [9] utilized ResNet50V2 and ArcFace to recognize sheep faces, and Wang et al. [11] used the YOLO algorithm for dairy cow face detection. 

Although the above methods have shown good potential performance, limitations still need to be addressed. Automatic sheep identification is regarded as the task of object detection and classification in most research. Consequently, as the key prerequisite for automatic sheep identification, the performance of sheep face detection is fateful to final identification. However, the performance of sheep face detection still cannot meet the requirements of actual sheep farms due to diverse face angles and scales, the high similarity of sheep faces, and so on, which hamper accurate sheep identification.

Early works have already made different efforts to improve the detection performance of sheep faces or body parts. Due to the fast development of human face detection algorithms (e.g., two-stage detection methods (R-CNN [12], Fast R-CNN [13]) and one-stage methods (SSD [14], YOLOv4 algorithm [15], RetinaNet [16], etc.), most of the previous research has usually applied the above methods to sheep face detection tasks. For example, U-Net and Network-II models were used to detect small-scale sheep faces [17]. Similarly, the Siamese network was used to detect and track the bodies of dairy goats [18]. As a typical two-stage human face detection algorithm, Faster R-CNN has also beenproven to be effective in detecting sheep and dairy goats [19,20]. In recent works, the popular detection YOLO series algorithm has been applied to sheep face detection. For example, Meng et al. [21] tried to use an improved YOLO framework to detect sheep faces. Moreover, the YOLOv4 algorithms have had good performance in detecting large-scale sheep face images [22]. 

Despite satisfactory results obtained by these intelligent detection methods based on human face detection technology, there are certain issues that still limit their practical application. Firstly, the diverse sheep face angle and scale in actual sheep farming have received little attention, and only a few studies have been conducted to construct an actual sheep dataset with characteristics of multi-scale and multi-location. Secondly, the high performance of sheep face detection often depends on complex deep learning models such as ResNet152, but it is accompanied by the problem of low efficiency, which seriously inhibits its practical applicability. RetinaFace, a single-stage detection algorithm widely used in human face detection, can obviously improve the accuracy of face identification [23]. The algorithm has the characteristic of multi-scale detection, which can be adapted to multi-scale detection in an actual farm environment. It can also predict the face box and face landmarks at the same time, which can support face alignment and effectively improves recognition accuracy. In this work, we propose an effective and lightweight sheep face detection method based on an improved RetinaFace algorithm. The proposed method uses improved MobileNetV3-large as the backbone network to extract multi-scale sheep facial features, and the single-stage headless detector (SSH) module is added with the channel and spatial attention mechanism to focus on important global facial features. Finally, the original multi-task loss function is optimized to fit the task of detecting sheep face box and facial landmarks. The experimental results on our collected real-world scenarios showed that the proposed method performs well in detecting multi-scale sheep faces in real-time, especially in the performance of detecting large-scale and small-scale sheep faces, which was obviously improved compared with other competing methods. In addition, we also demonstrate the effectiveness of the proposed method in an unseen environment by enabling zero-shot sheep face detection.

This paper aims to detect sheep faces and facial landmarks based on our improved RetinaFace algorithm, which can assist the breeder in identifying the ID of sheep quickly and accurately. Furthermore, this method is completely contactless and only uses the visual information of sheep’s face without any expensive auxiliary devices. Our main contribution can be summarized as: (1) we propose an improved RetinaFace algorithm to achieve accurate and real-time detection of sheep faces on actual sheep farms; (2) the improved MobileNetV3-large with a switchable atrous convolution (SAC) is introduced to accelerate the speed of multi-scale sheep face feature extraction; (3) the channel and spatial attention modules are added into the original detector module to obtain more discriminative sheep face features for detection; (4) a new sheep face dataset with characteristics of multi-scale have been conducted for open benchmark verification of sheep face detection.

The remaining paper is structured as follows. In Section 2, data collection, dataset construction, and the improved methods of the original algorithm are discussed. Experimental results discussion is shown in Section 3. In Section 4, the conclusions of this paper are summarized. It should be noted that the images of the dataset used in this paper are captured in an actual farm environment and included different scale faces of dairy goats and Ningxia Tan sheep. The code is available at https://github.com/haojy-2001/RetinaFace_SheepFaceDetectrion.git (accessed on 27 July 2023).

## 2. Materials and Methods

### 2.1. Dataset

The data used in this study are collected by recording two types of sheep: Ningxia Tan sheep and dairy goats. The rough videos of sheep faces are recorded by several cameras that are deployed at positions of different heights and angles on the ranch. The resolution of the videos is adjusted to 1920 × 1080 pixels.

After obtaining rough videos of sheep faces, some data pre-processing operations are adopted to construct the sheep image dataset used in this study. Firstly, to avoid data redundancy caused by the high similarity of the adjacent frames, the key frames are extracted from the videos by using the FFMpeg open software to obtain the sheep face images. Secondly, the face area and five landmarks of the sheep face image are manually annotated using the Labelme software. Some images with annotation information are shown in Figure 1. It should be noted that the collected dataset includes the face images with two types of background—inside and outside of the sheep house—to simulate the actual sheep ranch environment. Compared with the previous dataset [9], our new sheep face dataset has multi-scale characteristics, and the face region is located at different positions of the image, which provides a more challenging benchmark verification for sheep face detection and avoids overfitting during the training stage.

Overall, a total of 3749 images of 119 sheep were recorded in our new dataset. It includes 1415 images of 62 Ningxia Tan sheep and 2334 images of 57 dairy goats. According to the ratios of 6:2:2, the images of the dataset are randomly divided into the training set, the validation set, and the test set, which contains 2251, 748, and 750 images, respectively.

### 2.2. Sheep Face Detection Algorithm Based on the Improved RetinaFace

#### 2.2.1. Overview

In order to achieve real-time sheep face and facial landmarks detection on an actual sheep ranch, we propose a novel sheep face detection algorithm based on an improved RetinaFace. The structure of the proposed method is shown in Figure 2 and consists of three parts: an improved MobileNetV3-large network, feature pyramid networks (FPN), and an SHH-CBAM module. All of the input sheep face images were resized to 320 × 320 pixels. We used the same resize method (bilinear interpolation algorithm) as the original RetinaFace algorithm to avoid the problem of excessive model parameters in encoding high-resolution images. Given a resized image with a size of 320 × 320 pixels, the improved MobileNetV3-large network first takes it as an input to generate multi-resolution feature maps (here, the layers of C3, C4, and C5 are chosen). Then, the output feature maps of layer C3, C4, and C5 from the backbone network are respectively input into the FPN to obtain multi-scale features. In the FPN, the feature maps of C3, C4, and C5 are respectively processed by a series of operations of 1 × 1 convolution, up-sampling, elements addition, and 3 × 3 convolution [24], and finally, the multi-scale feature maps of C3’, C4’, and C5’ are obtained. Subsequently, in order to strengthen the ability of context modeling and to highlight important facial features of the sheep, the context module with CBAM takes feature maps of C3’, C4’, and C5’ as an input, respectively, and outputs the corresponding feature maps of P3, P4, and P5. Once the final multi-scale feature maps of P3, P4, and P5 are ready, the detector consisting of ClassHead, BoxHead, and LandmarkHead takes those final multi-scale feature maps as an input to predict the positions of the sheep faces and facial landmarks. The technical details above are presented in the sections below.

#### 2.2.2. Improved MobileNetV3-Large Network

Unlike the original RetinaFace algorithm that used complex deep learning models-ResNet50 as the backbone network, we propose a novel lightweight backbone network-improved MobileNetV3-large [25] with a switchable atrous convolution [26] to accelerate the speed of multi-scale sheep face feature extraction.

Table 1 provides the details of the proposed improved MobileNetV3-large network. As shown in the header of Table 1, Layers represent the building block of the network, and exp size is the channels of the feature maps after the expansion convolution. #out denotes the channels of the final output feature map. SE is used to indicate whether the building block uses the attention mechanism, and NL is the type of nonlinearity function (e.g., HS denotes the h-swish function, and RE is the ReLU function). s represents the stride of the convolution. It should be noted that the suffix symbol “(SAC)” represents our improved building blocks, where the depth-wise convolution is replaced by the SAC module (as a trainable expansion convolution), e.g., in the second row, the symbol of “bneck (SAC)“ represents that the depth-wise convolution of the original bneck block is replaced by SAC module with a stride of 2.

The structure of our bneck (SAC) is shown in Figure 3. The input feature maps of bneck (SAC) block are firstly processed by a 1 × 1 point-wise (Pwise) convolution and then followed by a 3 × 3 SAC module. In the SAC module, the input feature maps are operated by two global context modules and a switchable atrous convolution that is between those two global context modules. The input maps of the pre-global context module are first operated by global average pooling and a 1 × 1 convolution. Then the feature maps, after processing, are concatenated with the original input maps, which is regarded as the final output of the pre-global context module. Subsequently, the output feature maps are further input into the switchable atrous convolution. The processing procedure of the switchable atrous convolution S(x) is defined as:(1)$$S\left(x\right)=Conv\left(x,w_{1},1\right)\cdot f\left(x\right)+\left(1-f\left(x\right)\right)\cdot Conv\left(x,w_{2},3\right)$$
where *x* represents the input features, Conv(x, w, r) denotes a convolution layer with weight *w* and atrous rate *r*. f(x) represents the switchable function which consists of a 5 × 5 global average pooling layer and a 1 × 1 convolution. After obtaining the output of the switchable atrous convolution, we input it into the subsequent post-global context module whose structure was the same as the pre-global context module. After being processed by the SAC module, the feature maps will be further operated by an SE module and a 1 × 1 Pwise convolution. Finally, the processed feature map is added to the initial input map by the shortcut connection to generate the final output feature maps.

Given an image, the improved MobileNetV3 large network takes it as an input and outputs feature maps of C3, C4, and C5 layers, which are used as inputs for subsequent modules.

#### 2.2.3. SSH-CBAM Module with an Attention Mechanism

After obtaining the feature maps of C3, C4, and C5 layers, we employ the FPN module as the next step of our method to generate multi-scale features. More specifically, the C3, C4, and C5 feature maps are first processed by three 1 × 1 convolution layers, respectively. Then, the processed feature maps of C3 and C4 are merged with different upsampled maps (the processed C4 and the processed C5 are upsampled by a factor of 2) by element addition, respectively. Finally, the merged feature maps C3, C4, and C5 are further processed by three 3 × 3 convolution layers, respectively, to obtain the multi-scale feature maps C3’, C4’, and C5’.

Substantially, in order to avoid the problem of losing semantic information [27], we add the convolutional block attention module (CBAM) [28] into the context module of the original RetinaFace algorithm to increase the capability of extracting more common and important features. The original context module may lose some common features of sheep faces after integrating the multi-scale contextual semantics, while the CBAM module ensures the common sheep face features can be re-emphasized at different scales. The structure of our context + CBAM module is shown in Figure 4. When a new input feature map arrives, the det_conv1 branch (the upper route in Figure 4) first processes it by a 3 × 3 convolutional layer, followed by the CBAM module. For the det_context branch (the lower route Figure 4), it begins with a 3 × 3 convolutional layer and then separates into two branches. For the upper branch of det_context branch, the feature map is processed by a 3 × 3 convolutional layer and followed by a CBAM module. The lower branch of the det_context branch processes the feature map by two 3 × 3 convolutional layers and a CBAM module. The output feature maps of those three branches are concatenated together to generate the final feature map with context semantic information.

More specifically, the structure of the CBAM module is shown in Figure 5. It can be seen that the input feature maps are first processed by max-pooling and average-pooling, respectively. Then the outputs of the above two operations are input into a multi-layer perceptron (MLP) with a hidden layer. After that, the outputs of the MLP are merged by element-wise summation and then processed by the sigmoid function to obtain the tensor *M_C_* with enhanced channel features. *M_C_* is further processed by the element-wise multiplication with input feature maps to obtain the channel-refined feature maps. The channel-refined feature maps obtained by the CAM will be input into SAM.

In SAM, the channel-refined feature maps are operated by average-pooling and max-pooling along the channel axis. The generated feature maps are further operated by a 7 × 7 convolutional layer and a sigmoid function to obtain the tensor *M_S_* with enhanced spatial features. After that, *M_S_* is processed by element-wise multiplication with the channel-refined feature maps to obtain the final output. Overall, the channel-refined and spatial-refined feature maps can be obtained by adding a CBAM module.

After obtaining the multi-scale feature maps C3′, C4′, and C5′, the context + CBAM module takes them as an input, respectively, and outputs the corresponding feature maps of P3, P4, and P5. Once the final multi-scale feature maps of P3, P4, and P5 are ready, the detector consisting of ClassHead, BoxHead, and LandmarkHead takes those final multi-scale feature maps as input to predict the positions of sheep face and landmarks. According to the predicted results, we will finally get the predicted facial frames and landmarks by non-maximal suppression (NMS).

#### 2.2.4. Multi-Task Loss Function

The final multi-task loss function, including face classification loss, facial landmark regression loss, and box regression loss, is defined as:(2)$$L=L_{cls}\left(p_{i},p_{i}^{*}\right)+\lambda_{1}p_{i}^{*}L_{box}\left(t_{i},t_{i}^{*}\right)+\lambda_{2}p_{i}^{*}L_{pts}\left(l_{i},l_{i}^{*}\right)$$
where *i* represents any of the training anchor, $L_{cls}$ denotes the face classification loss function, which is the binary classification loss function based on the cross-entropy loss function. $P_{i}$ is the probability of anchor *i* containing a sheep face; when $p_{i}^{*}=1$, it indicates the positive anchor, otherwise, $p_{i}^{*}=0$ indicates the negative one. $L_{box}$ and $L_{pts}$ are the face box regression loss and the facial landmark regression loss, respectively, which are both Smooth-L1 functions. For face box regression loss $L_{box}$, $t_{i}={\left\{t_{x},t_{y},t_{w},t_{h}\right\}}_{i}$ and $t_{i}^{*}={\left\{t_{x}^{*},t_{y}^{*},t_{w}^{*},t_{h}^{*}\right\}}_{i}$ denote the coordinates of the predicted box and ground-truth positive anchor. In facial landmarks regression loss $L_{pts}$, $l_{i}={\left\{l_{x1},l_{y1},\ldots ,l_{x5},l_{y5}\right\}}_{i}$ and $l_{i}^{*}={\left\{l_{x1}^{*},l_{y1}^{*},\ldots ,l_{x5}^{*},l_{y5}^{*}\right\}}_{i}^{*}$ represent the predicted five facial landmarks and the manual labeled ones associated with the positive anchor. In our experiments, the loss coefficients $\lambda_{1}$ and $\lambda_{2}$ are set to 0.3 and 0.1, respectively.

### 2.3. Evaluation Indicators

In this paper, four indicators, including $F_{1}\;score$, average precision (AP), average processing time, and model size, are used to evaluate the performance of the detection model. The equation of the $F_{1}\;score$ is defined as:(3)$$F_{1}\;score=\frac{2\times Precision\times Recall}{Precision+Recall}$$

In Equation (3), the range of the precision and the recall are both in [0, 1], and the equations of these two indicators are defined in Equations (4) and (5), respectively.
(4)$$Precision=\frac{TP}{TP+FP}$$
(5)$$Recall=\frac{TP}{TP+FN}$$

In Equations (4) and (5), *TP* represents the true positives whose *IoU* (intersection over union) is more than 0.5, and *FP* denotes the false positives whose *IoU* is less than 0.5. *FN* is the false negatives whose *IoU* equals to 0.

Another indicator AP is calculated as:(6)$$AP=\int_{0}^{1}P\left(r\right)dr$$

In Equation (6), *P* represents a function that takes recall *r* as the independent variable. AP is the definite integral of the function, in which the value of recall is from 0 to 1. 

Furthermore, since the improved sheep face detection algorithm is designed for real-time detection, it is necessary to evaluate the detection efficiency of the model under actual conditions. Model size, average processing time, and Parameters (Param) are three indicators to evaluate the degree of lightweight and real-time performance of the algorithm. 

### 2.4. Experimental Setup

The experiment is implemented on a workstation that was installed with an NVIDA TITAN RTX/PCle/SSE2 graphics card and Ubantu18.04.5 operating system. In our experiment, the initial learning rate was set to $1\times 10^{-3}$. The training epochs was set to 200, and the batch size was set to 32. In addition, the Adam optimizer was used in this study. The beta1, beta2, and epsilon parameters of the Adam optimizer were kept default settings of 0.9, 0.999, and $1\times 10^{-8}$, respectively. The input scale of the images was resized to 320 × 320 pixels during the training process.

## 3. Results and Discussion

### 3.1. Performance on Sheep Face Detection

In order to verify the effectiveness of the proposed method, we compared our proposed method (RetinaFace + MobileNetV3-large + SAC + CBAM) with several mainstream detecting algorithms (YOLOv4-tiny [29], YOLOv5s [30], and EfficentDet [31]). The comparison results are shown in Table 2. For fair comparison, all other comparison detectors shown in Table 2 were retrained on our sheep dataset. In this verification, we used “RetinaFace + ResNet50” to denote the method of RetinaFace with the backbone network ResNet50, “RetinaFace + MobileNetV3-large + SAC + CBAM” to denote the method of RetinaFace with the improved MobileNetV3-large as the backbone network and with the SSH-CBAM module and “RetinaFace + ResNet50 + SAC + CBAM” to denote the method of the RetinaFace with improved ResNet50 as the backbone network and with the SSH-CBAM module. The improved ResNet50 used a SAC module to replace all of 3 × 3 convolutions at the first two stages of ResNet50.

As shown in Table 2, compared with all of the lightweight detectors and the original RetinaFace + ResNet50, our RetinaFace + MobileNetV3-large + SAC + CBAM obtained the best $F_{1}\;score$ of 95.25% and the best AP score of 96.00%, largely outperforming YOLOv4-tiny, YOLOv5s, and EfficientDet, which demonstrates that our method significantly outperformed the mainstream lightweight algorithms in detecting multi-scale sheep faces. Meanwhile, the proposed method had the lowest processing time, smallest model size, and lowest parameters, which proved the efficiency of the proposed method. It demonstrates that the proposed method is more suitable for multi-scale and real-time sheep face detection. After utilizing the lightweight backbone network and several improvement strategies, the model size and the average processing time of the proposed method was only about 10% of the baseline work, the RetinaFace + ResNet50, and the detection performance was slightly improved. The comparison results proved that the proposed method, with its efficient improvement, achieved a better performance than the method with the original deeper backbone network. According to the study [32], the walking speed of sheep is about 0.85 m/s. Thus, the proposed method can meet the need for real-time detection in the actual environment and has the best detecting performance compared with the mainstream lightweight algorithms and the original RetinaFace + ResNet50. Moreover, RetinaFace + ResNet50 + SAC + CBAM obtained a higher $F_{1}\;score$ and AP score compared with RetinaFace + ResNet50, which proves that our methods are still effective in the deep backbone network and can help the overall network achieve a significantly better detection performance. Compared with RetinaFace + ResNet50 + SAC + CBAM, our method (RetinaFace + MobileNetV3-large + SAC + CBAM) achieved a highly competitive accuracy performance with obviouslyless processing time and model size. On the whole, our method (RetinaFace + MobileNetV3-large + SAC + CBAM) outperformed the other state-of-the-art algorithms in detection performance by a large margin and achieved a competitive detection speed.

### 3.2. Qualitative Evaluation

In this evaluation, we choose the top four detection algorithms with the highest $F_{1}\;score$ (excluding RetinaFace + ResNet50 + SAC + CBAM): RetinaFace + MobileNetV3-large + SAC + CBAM, RetinaFace + ResNet50, YOLOv4-tiny, and YOLOv5s, to display the detection bounding boxes of those approaches. The qualitative comparisons between the top detection algorithms are shown in Figure 6. The detection bounding boxes of YOLOv4-tiny, YOLOv5s, and RetinaFace + ResNet50 were significantly reduced compared with the annotation boxes, as shown in the first line of Figure 6. The reason is that those methods cannot achieve multi-scale features, and they are especially weak in obtaining global face features from large-scale sheep faces. Meanwhile, YOLOv4-tiny failed to detect some small-scale sheep faces, as shown in the third line of Figure 6. In comparison with other approaches, the detection boxes of RetinaFace + MobileNetV3-large + SAC + CBAM almost overlapped the manual annotation boxes, which proved that the improved MobileNetV3-large network and SSH-CBAM module was helpful to obtain the global features of sheep faces. Furthermore, the RetinaFace + ResNet50 had a weak performance in detecting small-scale sheep faces, which proved that it could not effectively capture the key features of sheep faces. Overall, the proposed method obtained the best performance and was robust to detect all scales of sheep faces.

### 3.3. Sheep Face Detection under Different Light Conditions

In order to verify the effectiveness of the proposed method in different light conditions, we performed gamma transformation on the images of the test set to simulate the different light situations. We set γ=0.5 to simulate the illumination condition of 11:00 a.m. to 14:00 p.m., γ=1.5 to simulate the condition of 6:00 a.m. to 8:00 a.m. and 14:00 p.m. to 16:00 p.m., γ=2 to simulate the condition of 16:00 p.m. to 18:00 p.m., γ=3 to simulate the condition of 18:00 p.m. to 19:00 p.m. and 5:00 a.m. to 6:00 a.m., γ=4 to simulate the condition of 19:00 p.m. to 21:00 p.m.

The detection results under different light conditions are shown in Table 3. The proposed method still had a good performance in the range of γ from 0.5 to 2.0. When γ equalled 3.0 or 4.0, the recall rate of the proposed method declined to some extent, since those night data are extremely dark and even difficult for people to recognize.

The qualitative detection results under different light conditions are shown in Figure 7. The proposed method had a good performance in the range of γ from 0.5 to 2.0. When γ equalled 3.0, the detection boxes deviated from the annotation boxes slightly. When the γ equalled 4.0, there was no sheep face detected from the second image. The results indicate that as the lighting conditions decrease, the performance of the method will slightly decrease, but the detector is robust during the daytime period. The proposed method still needs to be improved when detecting faces in low illumination environments, such as in the sheep house at night. 

### 3.4. Ablation Experiment

**(1)** Effectiveness of the Different Key Components of the Proposed Methods

For the purpose of comparing the impact of different key components on the detecting performance, we conducted the ablation experiment. In this experiment, we used “RetinaFace+MobileNetV3-large” to denote the method of RetinaFace with the original MobileNetV3-large as the backbone network and “RetinaFace + MobileNetV3-large + SAC” to denote the method of RetinaFace with the improved MobileNetV3-large as the backbone network. As shown in Table 4, the proposed method RetinaFace + MobileNetV3-large + SAC + CBAM) achieved a considerable performance gain in both $F_{1}\;score$ and AP, when compared to other variant approaches. Compared with RetinaFace + ResNet50, the detection performance of the lightweight RetinaFace + MobileNetV3-large was decreased significantly in both $F_{1}\;score$ and AP, which demonstrated the lightweight backbone network weakened the feature extraction ability of the method. It can be observed that adding the SAC module could further improve the detection performance. The main reasons could be attributed to adding the SAC module into the C1–C2 layers of the MobileNetV3-large, helping to expand the receptive field, thus extracting large-scale sheep facial features from the images for sheep face detection. Moreover, our RetinaFace + MobileNetV3-large + SAC + CBAM outperformed the RetinaFace + MobileNetV3-large + SAC in terms of $F_{1}\;score$ and AP score, which demonstrated the effectiveness of the proposed SSH-CBAM module for sheep face detection. It can be noted that although adding SAC or CBAM brings a few additional computational costs, the proposed method (RetinaFace + MobileNetV3-large + SAC + CBAM) achieved considerable performance gains in both $F_{1}\;score$ and AP when compared to other baseline approaches.

**(2)** Effectiveness of the proposed detection method on the sheep face recognition task

As a key prerequisite, sheep face detection plays a crucial role in robust sheep face recognition. In order to verify the effect of sheep face detection methods on the performance of sheep face recognition, we conducted experiments on the publicly available sheep face recognition method (i.e., ArcFace [33]) based on the detection results of the detection methods (i.e., YOLOv4-tiny) and our proposed detection method). Specifically, we used the detection results as training data and test data for the sheep face recognition method.

The experimental results are shown in Table 5. Here, we used “YOLOv4-tiny + ArcFace” to denote the recognition method based on the detection results of YOLOv4-tiny, and “Ours + ArcFace” to denote the recognition method based on the detection results of our proposed detection method. As shown in Table 4, “Ours + ArcFace obtained the best recognition accuracy of 87.74%, outperforming YOLOv4-tiny + ArcFace by 16.31 percentage points. The experimental results proved that our sheep face detection method could boost the performance of a state-of-the-art publicly available sheep face recognition method.

**(3)** Ablation study of adding the SAC module into MobileNetV3-large

This study provided experiments for our improved MobileNetV3-large network that added the SAC module. We explored several choices for injecting such a SAC module within the original MobileNetV3-large network. The comparative results of adding the SAC module in different stages of the original MobileNetV3-large are shown in Table 6. It can be seen that adding the SAC module into deeper layers of MobileNetV3-large (e.g., C5, C4, and C3) significantly led to the decline in precision and recall scores, and the MobileNetV3-large + SAC_C2 achieved the best performance compared to other choices. It indicates that applying the SAC module in small-resolution feature maps (deeper layers) will lose part of the feature information, and the networks cannot capture the key facial features of Sheep. However, utilizing the SAC module at the appropriate position can effectively help the network to gain a larger receptive field and capture large-scale facial features. These results motivate us to use a MobileNetV3-large + SAC_C2 for the backbone network of our final architecture.

## 4. Conclusions

In this paper, we propose an improved lightweight RetinaFace algorithm for multi-scale sheep face detection. To accelerate the speed of the multi-scale sheep face feature extraction, an improved MobileNetV3-large with a switchable atrous convolution (SAC) was optimally used as the backbone network of the proposed algorithm. The channel and spatial attention modules were added into the original detector module to highlight important facial features of sheep. Furthermore, this work was trained and tested on a new sheep face dataset captured in a real sheep ranch with the characteristics of multi-scale. The experimental results on our new dataset demonstrate the effectiveness and robustness of the proposed method, which can be adopted to the multi-scale sheep face detection task in real time. It should be noted that the proposed method can be used in domains other than sheep face detection (e.g., cattle face detection or body detection of other animals), but these need to be further verified.

In the future, we will continue to explore an ensembled framework to properly integrate the improved sheep face detection model and the improved sheep face recognition model for identifying the IDs of the sheep quickly and accurately.

## Figures and Tables

**Figure 1 animals-13-02458-f001:**
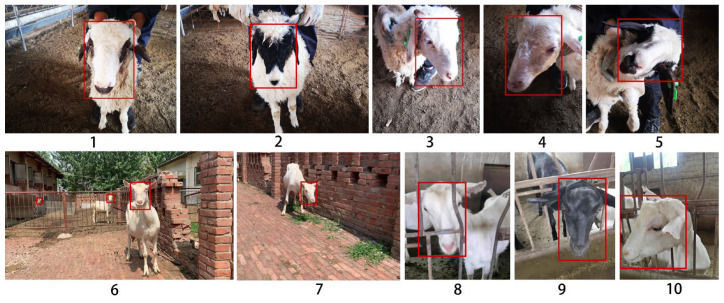
Some selected and annotated images of sheep faces from our dataset (1–5 for Ningxia Tan sheep, 6–10 for dairy goats).

**Figure 2 animals-13-02458-f002:**
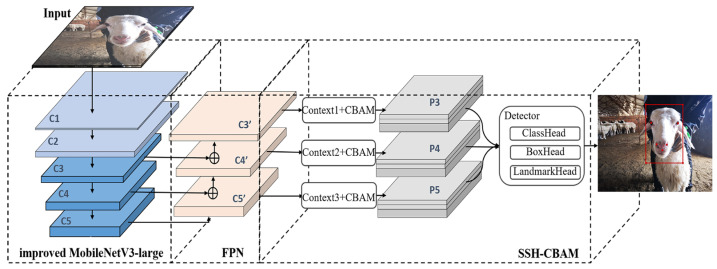
The architecture of the improved RetinaFace algorithm. The improved MobileNetV3-large network first takes a resized image as an input. Then, the output feature maps from the backbone network are respectively input into the FPN. The multi-scale feature maps of C3’, C4’, and C5’ are further input into the SSH-CBAM module to obtain the prediction results.

**Figure 3 animals-13-02458-f003:**
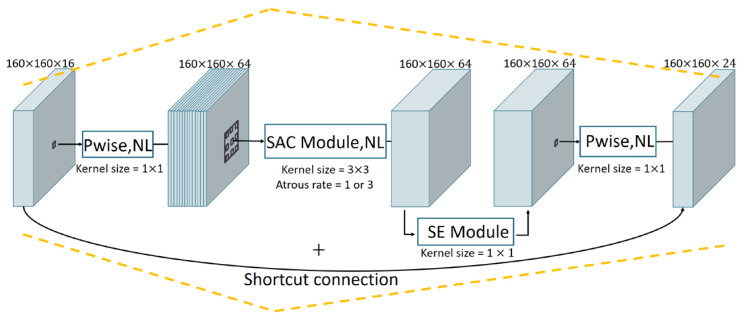
The bneck (SAC) structure of the C1 layer in the improved MobileNetV3-large. The input feature maps are processed by the Pwise module, SAC module, SE module, and Pwise module, sequentially. The output feature maps of the last Pwise module are added with the initial input map by the shortcut connection as the bneck module’s final output.

**Figure 4 animals-13-02458-f004:**
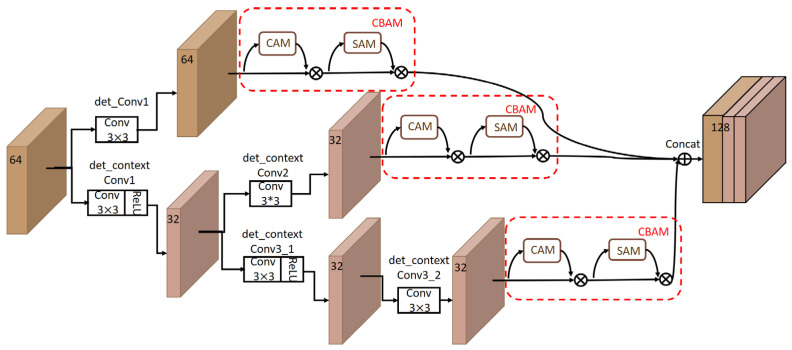
The improved context module with a CBAM. The input feature maps are processed by the det_conv1 branch and det_context branch, respectively. In the det_context branch, the feature maps are further separated into two branches to extract the context semantics. Finally, the output feature maps of the three branches are input into a CBAM and concatenated together.

**Figure 5 animals-13-02458-f005:**
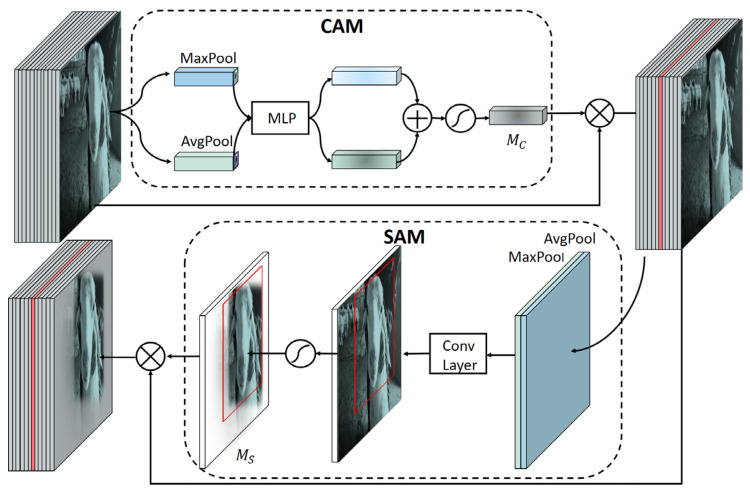
The structure of the CBAM module. MLP denotes a multi-layer perceptron (MLP) with one hidden layer. The input feature maps are processed by the CAM and SAM, sequentially. *M_C_* is the output from the CAM, which is element-wise multiplied with the original input to obtain the channel-refined feature maps. *M_S_* is the output from the SAM, which is element-wise multiplied with the channel-refined feature maps to obtain the final refined feature maps.

**Figure 6 animals-13-02458-f006:**
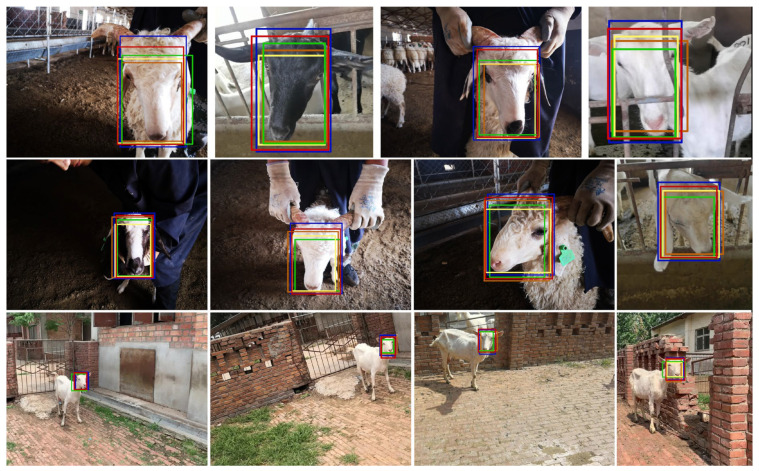
Qualitative results of different algorithms for sheep face detection. The blue box denotes the manual annotation face. The red box denotes the detection box of RetinaFace + MobileNetV3-large + SAC + CBAM, and the green one is the detection box of RetinaFace + ResNet50. The orange one denotes the detection box of YOLOv5s. The yellow one denotes the detection box of YOLOv4-tiny.

**Figure 7 animals-13-02458-f007:**
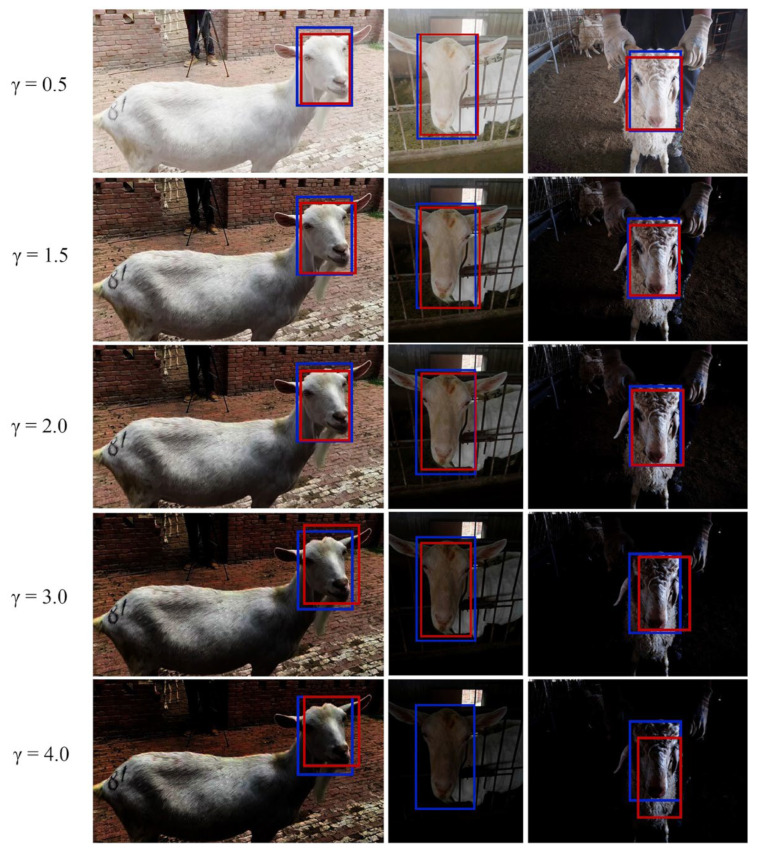
Qualitative detection results under different light conditions. The blue box denotes the manual annotation face. The red box denotes the detection box of the proposed method.

**Table 1 animals-13-02458-t001:** The improved MobileNetV3-large network structure. conv2d denotes 2D convolution layer. SE denotes whether there is a Squeeze and Excite in that block. NL denotes the type of nonlinearity used (e.g., HS denotes the h-swish function, and RE is the ReLU function). s denotes the stride of convolution. bneck denotes the bottleneck layer. NBN denotes no batch normalization. k denotes the user-defined channel size of the final output, where we have maintained the same value as the original MobileNetV3-large. SAC denotes whether there is a switchable atrous convolution replacing the original convolution layer.

Layers	Exp Size	#out	SE	NL	s
conv2d (SAC), 3 × 3	-	16	-	HS	2
Bneck (SAC), 3 × 3	16	16	-	RE	1
C1 bneck (SAC), 3 × 3	64	24	-	RE	1
bneck (SAC), 3 × 3	72	24	-	RE	1
C2 bneck (SAC), 3 × 3	72	40	✓	RE	2
bneck, 5 × 5	120	40	✓	RE	1
bneck, 5 × 5	120	40	✓	RE	1
C3 bneck, 3 × 3	240	80	-	HS	2
bneck, 3 × 3	200	80	-	HS	1
bneck, 3 × 3	184	80	-	HS	1
bneck, 3 × 3	184	80	-	HS	1
bneck, 3 × 3	480	112	✓	HS	1
bneck, 3 × 3	672	112	✓	HS	1
C4 bneck, 5 × 5	672	160	✓	HS	2
bneck, 5 × 5	960	160	✓	HS	1
bneck, 5 × 5	960	160	✓	HS	2
C5 conv2d, 1 × 1	-	960	-	HS	1
Pool, 7 × 7	-	-	-	HS	1
conv2d, 1 × 1, NBN	-	1280	-	-	1
conv2d, 1 × 1, NBN	-	k	-	HS	1

**Table 2 animals-13-02458-t002:** Comparisons of sheep face detection results on the proposed dataset.

Method	*F*_1_ *Score*	*AP*	Model Size	Average Processing Time	Param
YOLOv4-tiny [29]	77.05%	82.27%	23.56 MB	27.46 ms	5.87 M
YOLOv5s [30]	77.16%	95.19%	28.50 MB	27.22 ms	7.07 M
EfficentDet [31]	67.90%	73.40%	15.78 MB	42.63 ms	3.83 M
RetinaFace + ResNet50 [23]	94.02%	95.23%	119.02 MB	225.91 ms	29.67 M
RetinaFace + ResNet50 + SAC + CBAM	97.87%	96.44%	125.51 MB	266.62 ms	29.86 M
RetinaFace + MobileNetV3-large + SAC + CBAM	95.25%	96.00%	13.20 MB	26.83 ms	3.20 M

**Table 3 animals-13-02458-t003:** Comparisons of sheep face detection results for different γ.

γ	Recall	Precision	*F*_1_ *Score*
0.5	88.36%	95.65%	91.86%
1.5	91.95%	97.34%	94.57%
2.0	88.49%	96.41%	92.28%
3.0	76.33%	95.32%	84.77%
4.0	57.45%	94.21%	71.37%

**Table 4 animals-13-02458-t004:** Results of the ablation experiment.

Method	*F*_1_ *Score*	*AP*	Model Size	Average Processing Time
RetinaFace + ResNet50 [23]	94.02%	95.23%	119.02 MB	225.91 ms
RetinaFace + MobileNetV3-large	93.76%	89.83%	12.89 MB	22.32 ms
RetinaFace + MobileNetV3-large + SAC	95.12%	94.77%	13.19 MB	25.57 ms
RetinaFace + MobileNetV3-large + SAC + CBAM	95.25%	96.00%	13.20 MB	26.83 ms

**Table 5 animals-13-02458-t005:** Comparisons of sheep face recognition based on different detection methods.

Method	Accuracy
YOLOv4-tiny [29] + ArcFace [33]	71.43%
Ours + ArcFace [33]	87.74%

**Table 6 animals-13-02458-t006:** Sheep face detection results based on different improvement methods of MobileNetV3-large.

Method	Recall	Precision	*F*_1_ *Score*
MobileNetV3-large	90.31%	97.49%	93.76%
MobileNetV3-large + SAC_C5	63.10%	67.81%	65.37%
MobileNetV3-large + SAC_C4	72.86%	77.65%	75.18%
MobileNetV3-large + SAC_C3	86.70%	94.66%	90.50%
MobileNetV3-large + SAC_C2 (proposal)	92.22%	98.20%	95.12%
MobileNetV3-large + SAC_C1	91.33%	97.43%	94.28%

## Data Availability

The data presented in this study are available upon request from the corresponding author. The data are not publicly available due to the privacy policy of the authors’ institution.

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
