# Peer review of "Sheep Face Detection Based on an Improved RetinaFace Algorithm"

_animals, 2023, doi:10.3390/ani13152458_

Round 1

Reviewer 1 Report

Table 2 compares the paper's work with other widely used alternative face detectors, such as YOLO and RetinaFace. Have those other detectors been retrained on your sheep dataset? If not, then of course it is clear that your proposed detector yields much better results.

I think it would be better for the paper to make it totally clear in the chapter whether you compare (human) face detectors with your (sheep) detector, or whether they have also been trained with the sheep data. 

I found a few typos:

"propsed" instead of "proposed" (p 9)

"abaltaion" instead of "ablation" (p 12)

Author Response

Great thanks to you for your valuable efforts and constructive comments spent on our manuscript to make our work in very good shape. We also have put in our best efforts to reply all concerns, suggestions and questions raised by you as well as revise the manuscript accordingly.

Response to Reviewer 1 Comments

Great thanks to the Associate Editor and all the Reviewers for their valuable efforts and constructive comments spent on our manuscript to make our work in very good shape. We also have put in our best efforts to reply all concerns, suggestions and questions raised by all reviewers as well as revise the manuscript accordingly. We are sure that this process has helped us to improve the technical quality and the presentation of the revised manuscript substantially. Detailed responses to all comments are as follows.

Point 1: Table 2 compares the paper's work with other widely used alternative face detectors, such as YOLO and RetinaFace. Have those other detectors been retrained on your sheep dataset? If not, then of course it is clear that your proposed detector yields much better results. I think it would be better for the paper to make it totally clear in the chapter whether you compare (human) face detectors with your (sheep) detector, or whether they have also been trained with the sheep data.

Response 1: Thank you for your valuable suggestion. For fair comparison, all of other comparison detectors shown in Table 2 has been retrained on our sheep dataset. As suggested by reviewer, a clearer statement have been added into the section 3.1 of the revised manuscript.

Point 2: I found a few typos: "propsed" instead of "proposed" (p 9) "abaltaion" instead of "ablation" (p 12)

Response 2: Thank you for denoting the errors in the manuscript. We have carefully checked the paper and revised the typos or grammar errors. Besides, we have revised some sentence to make the description clearer in this revision of current manuscript.

Reviewer 2 Report

The manuscript proposes a novel model for sheep-face detection using improved retinaface algorithm with +MobileNetV3-large+SAC+CBAM. The results show that the proposed method is both fast and accurate. I have a few questions and suggestions for the authors which would help to justify the importance of the method.

1, Does improved retina face just mean using MobileNetV3 instead of ResNet50? please explain this clearly in the manuscript.

2, In table 2, a new case should be included that is having SAC and CBAM with the RetinaFace+ResNet50. This will show if the improvements in accuracy are achieved by the lightweight model or from SAC+CBAM. Also it justifies the method if the same accuracy is achieved with much less processing time.

3, In table 4, include the average processing time as well. It will help readers to understand which modules improve accuracy and their corresponding computational cost.

4, Please explain how the sheep face segments are resized to 320x320. Since the cropped face segments are rectangular, how can it be converted to a square image? Is it cropping or anisometric scaling? Please explain this in the manuscript and also explain why the respective choice is made.

Author Response

Great thanks to you for your valuable efforts and constructive comments spent on our manuscript to make our work in very good shape. We also have put in our best efforts to reply all concerns, suggestions and questions raised by you as well as revise the manuscript accordingly.

Response to Reviewer 2 Comments

Great thanks to the Associate Editor and all the Reviewers for their valuable efforts and constructive comments spent on our manuscript to make our work in very good shape. We also have put in our best efforts to reply all concerns, suggestions and questions raised by all reviewers as well as revise the manuscript accordingly. We are sure that this process has helped us to improve the technical quality and the presentation of the revised manuscript substantially. Detailed responses to all comments are as follows.

General Comment: The manuscript proposes a novel model for sheep-face detection using improved retinaface algorithm with +MobileNetV3-large+SAC+CBAM. The results show that the proposed method is both fast and accurate. I have a few questions and suggestions for the authors which would help to justify the importance of the method.

Response: Thank you for your valuable comments. We have revised the manuscript as suggested by the reviewer. Besides, we have revised some sentence to make the description clearer in this revision of current manuscript.

Point 1: Does improved retina face just mean using MobileNetV3 instead of ResNet50? please explain this clearly in the manuscript.

Response 1: Thank you for your question. Our improved retinaface has augmented the previous retinaface in different aspects. First, to accelerate the speed of multi-scale sheep face feature extraction, the improved MobileNetV3-large with Switchable Atrous Convolution (MobileNetV3-large+SAC) is optimally used to replace ResNet50 as the backbone network of the proposed algorithm. Second, the channel and spatial attention modules are added into detector module of original retinaface to highlight important facial features of sheep. This helps obtain more discriminative sheep face features to against the challenges of diverse face angles and scales in sheep. As suggested by the reviewer, we have added a paragraph at the end of the section 1 to make the description clearer in this revision of current manuscript.

Point 2: In table 2, a new case should be included that is having SAC and CBAM with the RetinaFace+ResNet50. This will show if the improvements in accuracy are achieved by the lightweight model or from SAC+CBAM. Also it justifies the method if the same accuracy is achieved with much less processing time.

Response 2: Thank you for your question. As suggested by the reviewer, we have added the experimental results of RetinaFace+ResNet50+SAC+CBAM into Table 2. As shown in Table 2, compared with RetinaFace+ResNet50+SAC+CBAM, our method (RetinaFace+MobileNetV3-large+SAC+CBAM) achieved highly competitive accuracy performance with more less processing time and model size. The corresponding result analysis mentioned above have been added into section 3.1.

Point 3: In table 4, include the average processing time as well. It will help readers to understand which modules improve accuracy and their corresponding computational cost.

Response 3: Thank you for your valuable comments. We have added the experimental results under the metrics of Model size and Average processing time in Table 4 of the revised manuscript. The corresponding analysis of experimental results is added at the end of first paragraph in section 3.4.

Point 4: Please explain how the sheep face segments are resized to 320x320. Since the cropped face segments are rectangular, how can it be converted to a square image? Is it cropping or anisometric scaling? Please explain this in the manuscript and also explain why the respective choice is made.

Response 4: Thank you for your valuable suggestion. The images are resized to 320×320 by using bilinear interpolation algorithm. We use the same resizing method as the original RetinaFace algorithm to avoid the problem of excessive model parameters in encoding high-resolution images. As suggested by reviewer, a clearer statement have been added into the section 2.2.1 of the revised manuscript.

Reviewer 3 Report

This is an interesting paper and the performance improvements described in your paper are notable.  Your improved algorithm can be used in domains other than sheep face detection – you should mention that in your conclusions.  Will you be making your dataset and code available to others?  Please provide links in your paper.

MobileNetV3 plays an important role in your research, but the MobileNetV3 article is not cited in your paper.  Please add a citation to "Searching for MobileNetV3" by Howard, et al.

Lines 174-183, describing the contents of Table 1, would work better if the information was placed inside the caption of Table 1, as was done in the Howard, et al. paper. Explain the acronyms SE = "Squeeze & Excite", NBN = "No Batch Normalization", and bneck = "bottleneck layer".

The captions for Figures 2- 5 do not describe the figures in enough detail to understand the figures without reading the text.  Please provide more details in the captions.

Figure 3 is not very informative.  Can you improve this figure?  At a minimum, perhaps you can include dimensions on the image layers to give the reader an idea of how the functional blocks are modifying the data.

Lines 220-229: Can you explain why you chose the specific architecture shown in Figure 4 for your context + CBAM module?

In Table 2, is model size measured in bytes? If so, please change the units to MB.  If not, please explain what the unit of measurement is.

Line 401: You state your algorithm outperforms YOLO by "16.31%" – this should be "16.31 percentage points".

Your paper could benefit from close editing by a native English speaker.

Author Response

Great thanks to you for your valuable efforts and constructive comments spent on our manuscript to make our work in very good shape. We also have put in our best efforts to reply all concerns, suggestions and questions raised by you as well as revise the manuscript accordingly.

Response to Reviewer 3 Comments

Great thanks to the Associate Editor and all the Reviewers for their valuable efforts and constructive comments spent on our manuscript to make our work in very good shape. We also have put in our best efforts to reply all concerns, suggestions and questions raised by all reviewers as well as revise the manuscript accordingly. We are sure that this process has helped us to improve the technical quality and the presentation of the revised manuscript substantially. Detailed responses to all comments are as follows.

Point 1: This is an interesting paper and the performance improvements described in your paper are notable. Your improved algorithm can be used in domains other than sheep face detection – you should mention that in your conclusions. Will you be making your dataset and code available to others? Please provide links in your paper.

Response 1: Thank you for your suggestion. We have re-considered our conclusion. The code of our method presented in this study are available in the link: https://github.com/haojy-2001/RetinaFace_SheepFaceDetectrion.git, we also put the link in the Introduction of our manuscript.

We also regret to inform you that the data are not publicly available due to the privacy policy of our institution.

Point 2: MobileNetV3 plays an important role in your research, but the MobileNetV3 article is not cited in your paper. Please add a citation to "Searching for MobileNetV3" by Howard, et al.

Response 2: Thank you for your valuable suggestion and reminder. We have added the relative citation at the first paragraph in Section 2.2.2.

Point 3: Lines 174-183, describing the contents of Table 1, would work better if the information was placed inside the caption of Table 1, as was done in the Howard, et al. paper. Explain the acronyms SE = "Squeeze & Excite", NBN = "No Batch Normalization", and bneck = "bottleneck layer".

Response 3: Thank you for your valuable suggestion. We have added the explanation information in the caption of Table 1, and explained the acronyms in detail.

Point 4: The captions for Figures 2-5 do not describe the figures in enough detail to understand the figures without reading the text.  Please provide more details in the captions.

Response 4: Thank you for your valuable comments. As suggested by the reviewer, we have provided more details in the captions for Figures 2-5, such as the summary of calculation process. We think your comments has greatly improved the quality of our manuscript.

Point 5: Figure 3 is not very informative. Can you improve this figure? At a minimum, perhaps you can include dimensions on the image layers to give the reader an idea of how the functional blocks are modifying the data.

Response 5: Thank you for your valuable comments. We have re-designed our Figure 3, which contains more details about the calculation procedure. We have added the dimensions of the feature maps and the operation of different layers in the Figure 3.

Point 6: Lines 220-229: Can you explain why you chose the specific architecture shown in Figure 4 for your context + CBAM module?

Response 6: Thank you for your valuable comments. We added an explanation of the reason why we chose the specific architecture of our context+CBAM module in Section 2.2.3.

Point 7: In Table 2, is model size measured in bytes? If so, please change the units to MB. If not, please explain what the unit of measurement is.

Response 7: Thank you for your question and reminder. We have changed the units to MB. Your comment really helps us to improve our manuscript.

Point 8: Line 401: You state your algorithm outperforms YOLO by "16.31%" – this should be "16.31 percentage points".

Response 8: Thank you for your valuable reminder. We have changed the statement. Thank you again for your valuable comment.

Point 9: Your paper could benefit from close editing by a native English speaker.

Response 9: Thank you for your suggestion. We have re-organized the expressions in the manuscript to conform the native English speaker expression, and we re-verified the grammar and wording of the entire text.

Round 2

Reviewer 2 Report

I am happy with the corrections made on the manuscript.